# FM-EAC: Feature Model-based Enhanced Actor-Critic for Multi-Task Control in Dynamic Environments

## Abstract

Model-based reinforcement learning (MBRL) and model-free reinforcement learning (MFRL) evolve along distinct paths but converge in the design of Dyna-Q Sutton & Barto (2018). However, modern RL methods still struggle with effective transferability across tasks and scenarios. Motivated by this limitation, we propose a generalized algorithm, FM-EAC, that *integrates planning, acting, and learning* for multi-task control in dynamic environments. FM-EAC combines the strengths of MBRL and MFRL and improves generalizability through the use of novel feature-based models and an enhanced actor-critic framework. Simulations in both urban and agricultural applications demonstrate that FM-EAC consistently outperforms many state-of-the-art MBRL and MFRL methods. More importantly, different sub-networks can be customized within FM-EAC according to user-specific requirements.

## 1 Introduction

Over the past few decades, it has been a highly debated topic of what the best approach is for decision-making, i.e., via planning or learning. Within a Markov decision process (MDP), where one or multiple agents interact with the environment, experience serves at least two key roles. First, it can be used to improve the model so that it can more accurately reflect the real environment - this is known as planning or model-based reinforcement learning (MBRL). Second, experience can be used directly to enhance the value function and policy - this is known as model-free reinforcement learning (MFRL) Sutton & Barto (2018).

On the one hand, MBRL methods, including MBPO Janner et al. (2019), MORel Kidambi et al. (2020), COMBO Yu et al. (2021), CMBAC Wang et al. (2022), and Dreamer Hafner et al. (2025), make use of a limited amount of experience and thus achieve a higher training efficiency with fewer environmental interactions. On the other hand, MFRL methods, including DQN Mnih et al. (2013), DDPG Lillicrap et al. (2016), PPO Schulman et al. (2017), SAC Haarnoja et al. (2018), and TD3 Dankwa & Zheng (2020), are much simpler and are not affected by biases in the design of the model, thus more suitable for complex environments. Despite the development on these two approaches, there exist many similarities between MBRL and MFRL methods, and such insights are reflected in the design of Dyna-Q Sutton & Barto (2018), which combines model-free learning with simulated experience from a learned model.

However, a common limitation of existing MBRL and MFRL approaches lies in their lack of *transferability*. In the context of MBRL, a model of the environment (a.k.a., state transition model) means anything that an agent can use to predict how the environment will respond to its actions. Given a state and an action, a state transition model produces a prediction of the upcoming state and reward. Thus, MBRL methods are often tailored to specific models and environments. Unlike MBRL, MFRL methods utilize value and policy iterations to improve the optimality of the policies. The modern MFRL methods, especially those based on the actor-critic framework, perform well in complex tasks such as unmanned aerial vehicle (UAV) control Liu et al. (2024); Zhao et al. (2024). Nevertheless, they are typically trained in fixed environments for individual tasks, which highlights the need for a generalized method capable of handling different tasks in dynamic environments.

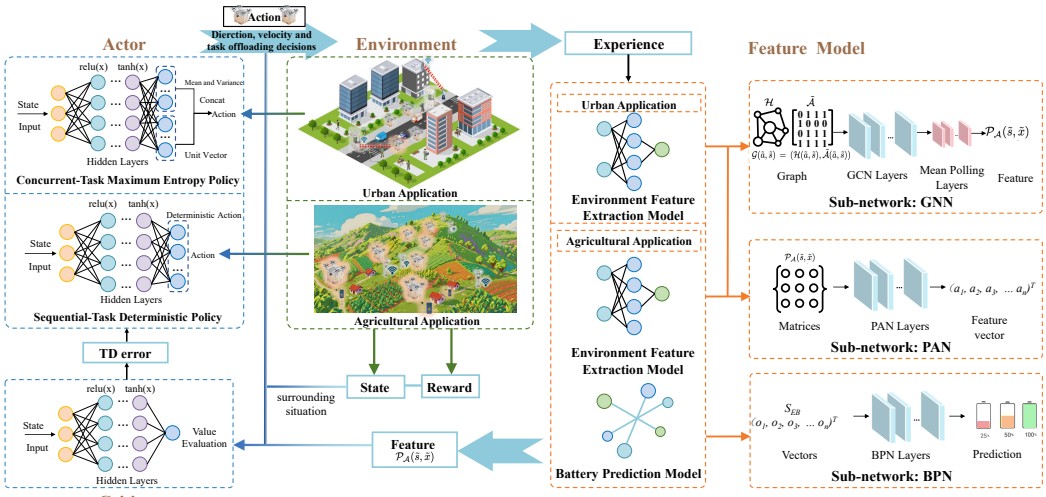

Figure 1: Overview of FM-EAC.

To this end, we envision a generalized model integrating planning, acting, and learning for multi-task control in dynamic environments. Our proposed algorithm, referred to as feature model-based enhanced actor-critic (FM-EAC), combines the advantages of MBRL and MFRL approaches while improving the generalizability and transferability. Real-world environments are featured by spatio-temporal variations. In this context, generalizability refers to the ability of the algorithm to adapt to various environmental conditions and task requirements, and transferability refers to the ability of the algorithm to remain robust when environmental conditions or task requirements change.

Beyond existing methods, FM-EAC has two distinct modules: a feature model and an enhanced actor-critic framework. Despite the spatiotemporal variations in dynamic environments, it is possible to extract environmental features as representations and leverage them to train a feature model that is robust to environmental uncertainties. Such a feature model is combined with the state to enrich environmental information. Meanwhile, the actor-critic framework in MFRL offers higher flexibility for multi-task setups. Therefore, we propose an enhanced actor-critic framework to decouple the actors and critics for different tasks, enabling simultaneous policy updates.

The main contributions are summarized below:

- FM-EAC, a feature model-based enhanced actor-critic algorithm, is proposed for multi-task control in dynamic environments. It combines the benefits of both MBRL and MFRL with improved generalizability and transferability.

- Within FM-EAC, we demonstrate three exemplary sub-networks: graph neural network (GNN) Scarselli et al. (2009), point array network (PAN), and battery prediction network (BPN). GNN and PAN are adaptively trained networks and pre-trained frozen networks, respectively, for feature extraction of environments. BPN is a pre-trained frozen network for capacity prediction of batteries.

- Different from existing reinforcement learning methods that are task- and environment-specific, our proposed FM-EAC can learn from various environments for multiple tasks, surpassing the state-of-the-art methods in performance, efficiency, and stability when environmental conditions or task requirements change.

## 2 OVERVIEW OF FM-EAC

FM-EAC, as illustrated in Fig. 1, is composed of three main components: (1) the environment, (2) the enhanced actor-critic framework, and (3) the feature model.

We demonstrate two exemplary environments for UAV multi-task control. The first is an urban application, where multiple UAVs are deployed for package delivery Betti Sorbelli (2024); Zieher

et al. (2024) and mobile edge computing (MEC) Zhou et al. (2020); Ning et al. (2023) for Internet of Things (IoT) devices, i.e., pedestrian-carrying devices (PDs) and ground devices (GDs). The other is an agricultural application Agrawal & Arafat (2024); Rejeb et al. (2022), where multiple UAVs are deployed for data collection from wireless sensors (WSs) and charging at docking stations (DSs) when needed.

To enhance generalization across scenarios, we follow a modular design for the enhanced actor-critic framework. Since the data collection and battery charging are sequential tasks (i.e., one objective at one time), we employ a deterministic actor-critic in the agricultural application. Meanwhile, the package delivery and MEC are concurrent tasks (i.e., multiple objectives at one time with a priority order); thus, we utilize a maximum entropy actor-critic in the urban application. More specifically, the actor network is modified accordingly to produce the primary maximum entropy action for the main task (e.g., UAV trajectory planning), while simultaneously generating task offloading decisions for MEC as a supplementary output.

Compared to conventional actor-critic methods, our FM-EAC leverages a novel feature model to generate scenario-related features, which function as specialized evaluation indicators for the critic's estimation of state-action values. Note that the structures of sub-networks in the feature model are highly flexible: GNN is a representation of the compute-intensive yet information-rich adaptive learning network, while PAN and BPN are representations of pre-trained and lightweight frozen networks; beyond the GNN, PAN, and BPN adopted in this paper, they can be substituted with other neural networks or even predefined feature matrices in user-defined scenarios.

Upon deployment, FM-EAC consists of two phases: (1) execution and (2) training, where the flows of both phases are shown in Fig. 1. In the execution phase, actor networks generate action policies, which are applied to the scenario environment to update states and receive corresponding rewards. Simultaneously, in the training phase, agents extract environmental experiences to construct environment feature models, thereby producing scenario-aware features. These features, along with actions, states, and rewards, are then utilized by the critic during execution to estimate state-action values and guide the update of the actor networks.

## 3 DESIGN OF FM-EAC

### 3.1 ENHANCED ACTOR-CRITIC FRAMEWORK

The enhanced actor network has two parts: a linear layer with output of distribution parameters $\boldsymbol{\mu}$ and $\boldsymbol{\sigma^2}$, and a softmax layer with output of a unit vector $\boldsymbol{\delta}_r$. Among them, $\boldsymbol{\mu}$ and $\boldsymbol{\sigma^2}$ are utilized for generating independent action variables for sequential tasks. Meanwhile, $\boldsymbol{\delta}_r$ is used for producing correlated action variables for concurrent tasks. Besides, the input of the critic network consists of observation, action, and environmental features $[\boldsymbol{o}, \boldsymbol{a}, \boldsymbol{f}_e]$, and the output is the evaluated value $V(\boldsymbol{o}, \boldsymbol{a}, \boldsymbol{f}_e)$.

As mentioned above, we consider two types of tasks: the sequential and concurrent tasks. Taking the concurrent tasks as an example, assuming that $\mathcal{A}_\mathcal{B}$, $\mathcal{S}_\mathcal{B}$, $\mathcal{R}_\mathcal{B}$, $\hat{\mathcal{R}}_\mathcal{B}$, and $\mathcal{S}'_\mathcal{B}$ represent the actions, states, primary task rewards, secondary task rewards, and next states from a batch size of sampled data, respectively. $\pi(\mathcal{A}_\mathcal{B}|\mathcal{S}_\mathcal{B})$ represents the policy or actor network. To achieve better training performance, four critic networks, $\mathcal{Q}_{\mathcal{P}1}(\mathcal{S}_\mathcal{B}, \mathcal{A}_\mathcal{B})$, $\mathcal{Q}_{\mathcal{P}2}(\mathcal{S}_\mathcal{B}, \mathcal{A}_\mathcal{B})$, $\mathcal{Q}_{\mathcal{S}1}(\mathcal{S}_\mathcal{B}, \mathcal{A}_\mathcal{B})$, and $\mathcal{Q}_{\mathcal{S}2}(\mathcal{S}_\mathcal{B}, \mathcal{A}_\mathcal{B})$ are applied. Among them, $\mathcal{Q}_{\mathcal{P}1}(\mathcal{S}_\mathcal{B}, \mathcal{A}_\mathcal{B})$ and $\mathcal{Q}_{\mathcal{P}2}(\mathcal{S}_\mathcal{B}, \mathcal{A}_\mathcal{B})$ are updated by primary task rewards, while $\mathcal{Q}_{\mathcal{S}1}(\mathcal{S}_\mathcal{B}, \mathcal{A}_\mathcal{B})$, and $\mathcal{Q}_{\mathcal{S}2}(\mathcal{S}_\mathcal{B}, \mathcal{A}_\mathcal{B})$ are updated by secondary task rewards. We also design the corresponding target critic networks: $\mathcal{Q}'_{\mathcal{P}1}(\mathcal{S}_\mathcal{B}, \mathcal{A}_\mathcal{B})$, $\mathcal{Q}'_{\mathcal{P}2}(\mathcal{S}_\mathcal{B}, \mathcal{A}_\mathcal{B})$, $\mathcal{Q}'_{\mathcal{S}1}(\mathcal{S}_\mathcal{B}, \mathcal{A}_\mathcal{B})$, and $\mathcal{Q}'_{\mathcal{S}2}(\mathcal{S}_\mathcal{B}, \mathcal{A}_\mathcal{B})$, which copy the parameters from critic networks periodically.

The enhanced actor-critic framework can be implemented via the following steps:

1. Initialize the update parameter for the actor network $\theta_A$, the update parameter for the critic networks $\phi_{P_1}$, $\phi_{P_2}$, $\phi_{S_1}$ and $\phi_{S_2}$, and the update parameters for the target critic networks $\phi'_{P1}$, $\phi'_{P2}$, $\phi'_{S1}$ and $\phi'_{S2}$. Initialize the replay buffer $\mathcal{D}$.

2. During an episode $epi$, let the agents interact with the environment and store the observation $\boldsymbol{o}_i$, action $\boldsymbol{a}_i$, primary task rewards $r_i$, and secondary task rewards $\hat{r}_i$ to the replay buffer $\mathcal{D}$.

3. Randomly sample one batch size data $\mathcal{A}_\mathcal{B}$, $\mathcal{S}_\mathcal{B}$, $\mathcal{R}_\mathcal{B}$, $\hat{\mathcal{R}}_\mathcal{B}$ and $\mathcal{S}'_\mathcal{B}$ from $\mathcal{D}$.

4. Use the target critic networks and the current policy to compute the target value. Specifically, the primary target Q-values $\mathcal{Y}_\mathcal{P}$ and secondary target Q-values $\mathcal{Y}_\mathcal{S}$ can be calculated using the minimum of the two target critics, following the Clipped Double Q-learning strategy to mitigate overestimation bias. The formulations are as follows:

$$\mathcal{Y}_\mathcal{P} = \mathcal{R}_\mathcal{B} + \gamma(1 - d_\mathcal{B}) \min\{\mathcal{Q}'_{\mathcal{P}1}(\mathcal{S}'_\mathcal{B}, \mathcal{A}'_\mathcal{B}), \mathcal{Q}'_{\mathcal{P}2}(\mathcal{S}'_\mathcal{B}, \mathcal{A}'_\mathcal{B})\}, \tag{1}$$

$$\mathcal{Y}_\mathcal{S} = \hat{\mathcal{R}}_\mathcal{B} + \gamma(1 - d_\mathcal{B}) \min\{\mathcal{Q}'_{\mathcal{S}1}(\mathcal{S}'_\mathcal{B}, \mathcal{A}'_\mathcal{B}), \mathcal{Q}'_{\mathcal{S}2}(\mathcal{S}'_\mathcal{B}, \mathcal{A}'_\mathcal{B})\}. \tag{2}$$

where $d_\mathcal{B}$ represents the termination flag.

5. Minimize the mean squared error (MSE) between the predicted Q-values and the target Q-values for critic update. We update the primary and secondary critic networks by minimizing the loss function $\mathcal{J}_{\mathcal{QP}i}(\phi_T)$ and $\mathcal{J}_{\mathcal{QS}i}(\phi_P)$:

$$\mathcal{J}_{\mathcal{QP}i}(\phi_P) = \mathbb{E}[(\mathcal{Q}_{\mathcal{P}i}(\mathcal{S}_\mathcal{B}, \mathcal{A}_\mathcal{B}) - \mathcal{Y}_\mathcal{P})^2], i \in \{1, 2\}, \tag{3}$$

$$\mathcal{J}_{\mathcal{QS}i}(\phi_S) = \mathbb{E}[(\mathcal{Q}_{\mathcal{S}i}(\mathcal{S}_\mathcal{B}, \mathcal{A}_\mathcal{B}) - \mathcal{Y}_\mathcal{S})^2], i \in \{1, 2\}, \tag{4}$$

where $\mathbb{E}(\cdot)$ represents the mathematical expectation.

6. Update actor network by maximizing the sum of the estimated Q-values from the primary critic $\mathcal{Q}_{\mathcal{P}1}(\mathcal{S}_\mathcal{B}, \mathcal{A}_\mathcal{B})$ and that of secondary critic $\mathcal{Q}_{\mathcal{S}1}(\mathcal{S}_\mathcal{B}, \mathcal{A}_\mathcal{B})$. For concurrent tasks based on maximum entropy policy, the loss function $\mathcal{J}_\pi(\theta_A)$ can be represented as:

$$\mathcal{J}_\pi(\theta_A) = \mathbb{E}[\mathcal{Q}_{\mathcal{P}1}(\mathcal{S}_\mathcal{B}, \mathcal{A}_\mathcal{B}) + \mathcal{Q}_{\mathcal{S}1}(\mathcal{S}_\mathcal{B}, \mathcal{A}_\mathcal{B}) - \alpha \log \pi_{\theta_A}(\mathcal{A}_\mathcal{B}|\mathcal{S}_\mathcal{B})], \tag{5}$$

where $\alpha$ represents temperature parameter.

For sequential tasks based on deterministic policy, the loss function can be represented as:

$$\mathcal{J}_\pi(\theta_A) = \mathbb{E}[\mathcal{Q}_{\mathcal{P}1}(\mathcal{S}_\mathcal{B}, \pi_{\theta_A}(\mathcal{A}_\mathcal{B}|\mathcal{S}_\mathcal{B})) + \mathcal{Q}_{\mathcal{S}1}(\mathcal{S}_\mathcal{B}, \pi_{\theta_A}(\mathcal{A}_\mathcal{B}|\mathcal{S}_\mathcal{B}))]. \tag{6}$$

7. Softly update target critic networks to ensure training stability. We slowly update the target critic networks toward the current critic networks. Instead of directly copying the weights, the target networks are updated using a weighted average of the current and previous target weights:

$$\phi'_{Pi} = \xi \phi_{P_i} + (1 - \xi)\phi'_{Pi}, i \in \{1, 2\}, \tag{7}$$

$$\phi'_{Si} = \xi \phi_{S_i} + (1 - \xi)\phi'_{Si}, i \in \{1, 2\}, \tag{8}$$

where $\xi$ is the soft update parameter.

8. Repeat step 2-7 in all the training iterations until policy converges.

Different from the conventional actor-critic framework, we divide the output layers of the actor and use a softmax to enable the actor to deal with different task action outputs. We use four different critics to evaluate the total task action value and the partial action value. In this way, the enhanced actor-critic framework retains the advantages of state-of-the-art actor-critic methods while enabling seamless decision-making during multi-task control.

## 3.2 GNN, PAN, AND BPN SUB-NETWORKS

The environment for UAV multi-task control is highly dynamic. The uncertainty comes from various aspects, including ground topology and elevation, building distributions and heights, locations of base stations (BSs), GDs, and WSs, the trajectory of PDs, and the origin and destinations of UAVs. Under these circumstances, we design three distinct sub-networks for feature extraction and prediction of environmental parameters, namely GNN, PAN, and BPN.

We also consider alternative approaches that utilize GNN and PAN for environmental feature extraction, referred to as GNN-EAC and PAN-EAC, respectively. GNN-EAC takes the graph relationships among UAVs and other entities as features. PAN-EAC takes pre-trained PAN as features.

**Algorithm 1** The training process of the GNN network.

1: Initialize network $\mathcal{F}_{\text{GNN}}(\varphi)$.
2: **for** $epi \in \{0, 1, \ldots, n\}$ **do**
3:    Reset environment.
4:    **for** $step \in \{0, 1, \ldots, m\}$ **do**
5:        Generate graph $\mathcal{G}(\tilde{a}, \tilde{s})$ as $(\mathcal{H}(\tilde{a}, \tilde{s}), \tilde{\mathcal{A}}(\tilde{a}, \tilde{s})$
6:        **Outputs**: $\mathcal{F}_{\text{GNN}}(\mathcal{G}(\tilde{a}, \tilde{s}))$ and Feature $\mathcal{F}_{\text{env}}$ as equation 11.
7:        Update $\mathcal{F}_{\text{GNN}}(\varphi)$ by equation 10.
8:    **end for**
9: **end for**

**Algorithm 2** The training process of the PAN network.

1: Initialize network $\mathcal{F}_{\text{PAN}}(\omega)$ and $\mathcal{F}_{SN}(\mathcal{F}_{\text{PAN}}(\omega)))$.
2: Generate a point array dataset $D_{\mathcal{P}_{\mathcal{A}}}$.
3: **for** $Epoch \in 0, 1, \ldots, k$ **do**
4:    Extract a point array $\mathcal{P}_{\mathcal{A}}(\tilde{s}, \tilde{x})$ from $D_{\mathcal{P}_{\mathcal{A}}}$.
5:    Update $\mathcal{F}_{\text{PAN}}(\omega)$ and $\mathcal{F}_{SN}(\mathcal{F}_{\text{PAN}}(\omega)))$ by equation 12.
6: **end for**
7: **Outputs**: Model $\mathcal{F}_{\text{PAN}}(\mathcal{P}_{\mathcal{A}}(\tilde{s}, \tilde{x}))$.
8: **for** $epi \in \{0, 1, \ldots, n\}$ **do**
9:    Reset environment.
10:    **for** $step \in \{0, 1, \ldots, m\}$ **do**
11:        Extract environmental point array $\mathcal{P}_{\mathcal{A}}(\tilde{s}_{\text{step}}, \tilde{x}_{\text{step}})$
12:        **Output**: Feature $\mathcal{F}_{\text{env}}(\tilde{s})$ as equation 13.
13:    **end for**
14: **end for**

GNN-EAC consists of graph convolutional network (GCN) layers and mean pooling layers. First, a graph structure $\mathcal{G}(\tilde{a}, \tilde{s}) = (\mathcal{H}(\tilde{a}, \tilde{s}), \tilde{\mathcal{A}}(\tilde{a}, \tilde{s}))$, which consists of node features $\mathcal{H}(\tilde{a}, \tilde{s})$ and adjacency matrix $\tilde{A}(\tilde{a}, \tilde{s})$, is designed, where $\tilde{a}$ represents the corresponding agent nodes, $\tilde{s}$ represents scenario feature nodes. $\mathcal{G}(\tilde{a}, \tilde{s})$ is utilized to describe the relationship between agents and corresponding factors. In the urban application, it consists of UAVs, BSs, GDs, and PDs. In the agricultural application, it consists of UAVs and WSs. Then, $\mathcal{G}(\tilde{a}, \tilde{s})$ is input to GCN layers. The output of the GCN layers can be denoted as:

$$\mathcal{H}(\tilde{a}, \tilde{s})^T = \varsigma \left( \tilde{D}^{-\frac{1}{2}} \tilde{\mathcal{A}} \tilde{D}^{-\frac{1}{2}} \mathcal{H}(\tilde{a}, \tilde{s}) W \right), \tag{9}$$

where $(\cdot)^T$ represents matrix transposition, $\tilde{D}$ represents degree matrix, $W$ represents the learnable parameter matrix, and $\varsigma$ represents the non-linear activation function, for which we have chosen ReLU in this work. After that, $\mathcal{H}(\tilde{a}, \tilde{s})^T$ is input to the mean pooling layers for output. This GNN network will be updated as follows:

$$\mathcal{J}_{\mathcal{F}\mathcal{G}}(\varphi) = \mathcal{J}_{\mathcal{Q}\mathcal{P}i}(\phi_P) + \mathcal{J}_{\mathcal{Q}\mathcal{S}i}(\phi_S) + \mathcal{J}_{\pi}(\theta_A), i \in 1, 2, \tag{10}$$

where $\varphi$ represents the updating parameter, $\mathcal{J}_{\mathcal{F}\mathcal{G}}(\varphi)$ represents the loss function, and $\mathcal{J}_{\mathcal{P}\mathcal{T}i}(\phi_P)$, $\mathcal{J}_{\mathcal{Q}\mathcal{S}i}(\phi_S)$, and $\mathcal{J}_{\pi}(\theta_A)$ are mentioned before. Finally, we get a GNN feature model $\mathcal{F}_{\text{GNN}}(\mathcal{G}(\tilde{a}, \tilde{s}))$. The environment features $\mathcal{F}_{\text{env}}$ can be represented as:

$$\mathcal{F}_{\text{env}} = \beta_G \cdot \mathcal{F}_{\text{GNN}}(\mathcal{G}(\tilde{a}, \tilde{s})), \tag{11}$$

where $\beta_G$ represents the feature normalization coefficient. The training process of GNN is shown in Algorithm 1.

In contrast, PAN-EAC utilizes the pre-trained PAN network to extract environment features. It relies on prior experience from the corresponding environment for decision-making to some extent, performing better in pre-known scenarios while maintaining a reliable predictive capability for new environments. First, we generate a point array dataset $\mathcal{P}_{\mathcal{A}}(\tilde{s}, \tilde{x})$, where $\tilde{x}$ represents feature indicator. In the urban application, it includes the GDs' and PDs' information, and in the agricultural application, it consists of WSs' information. Then we assume that $\tilde{S}$ and $\tilde{X}$ represent a batch of $\tilde{s}$ and $\tilde{x}$, respectively. $\tilde{S}$ and $\tilde{X}$, which are padded to the same dimension, are input to the PAN layers $\mathcal{F}_{\text{PAN}}(\cdot)$, and the output will be sent to a sequential network $\mathcal{F}_{SN}(\cdot))$ to obtain the prediction indicator $\tilde{X}'$. After that, the PAN layers and the connected sequential network will be trained as follows:

$$\mathcal{J}_{\mathcal{F}\mathcal{P}}(\omega) = ||\{\tilde{S}, \tilde{X}\} - \{\tilde{S}, \tilde{X}'\}||^2, \tag{12}$$

where $|| \cdot ||$ represents the Euclidean distance function, $\omega$ represents the updating parameter, and $\mathcal{J}_{\mathcal{F}\mathcal{P}}(\omega)$ represents the loss function for PAN network. Finally, we remove the sequential network and get a feature extraction model $\mathcal{F}_{\text{PAN}}(\mathcal{P}_{\mathcal{A}}(\tilde{s}, \tilde{x}))$, and the $\mathcal{F}_{\text{env}}$ can be represented as:

| **Algorithm 3** Training process of BPN. | **Algorithm 4** Training process of FM-EAC. |
|---|---|
| 1: Pre-train the model $\mathcal{Q}_{EM}(\mathcal{S}_{\mathcal{B}}, \mathcal{A}_{\mathcal{B}})$ according to the scenarios and task setting. | 1: Initialize network $\pi(\cdot)$, $\mathcal{Q}(\cdot)$, and $\mathcal{F}(\cdot)$. |
| 2: Initialize network $\mathcal{F}_{\text{BPN}}(\epsilon)$. | 2: Define network $\mathcal{F}_{\text{BPN}}(\cdot)$ according to the scenario. |
| 3: Generate dataset $\{\mathcal{S}_E, \mathcal{X}_E\}$ from environment interaction according the scenario setting. | 3: Pre-train $\mathcal{F}(\cdot)$ and $\mathcal{F}_{\text{BPN}}(\cdot)$ as **Algorithm 2** and **3**. |
| 4: **for** $Epoch \in \{0, 1, \ldots, k\}$ **do** | 4: **for** $epi \in \{0, 1, \ldots, n\}$ **do** |
| 5:    Extract a batch size $\{\mathcal{S}_{EB}, \mathcal{X}_{EB}\}$ from $\{\mathcal{S}_E, \mathcal{X}_E\}$. | 5:    Reset environment. |
| 6:    Update $\mathcal{F}_{\text{BPN}}(\epsilon)$ by equation 14. | 6:    **for** $step \in \{0, 1, \ldots, m\}$ **do** |
| 7: **end for** | 7:      Output action $\boldsymbol{a}_i$ from $\pi(\cdot)$. |
| 8: **Outputs**: Model $\mathcal{F}_{\text{BPN}}(s_E)$. | 8:      Interact with environment. |
| | 9:      Store $\boldsymbol{o}_i$, $\boldsymbol{a}_i$, $r_i$, $\hat{r}_i$, and $\boldsymbol{o}_{i+1}$ to $\mathcal{D}$. |
| | 10:     Update $\pi(\cdot)$ and $\mathcal{Q}(\cdot)$ by equation 1 to 8. |
| | 11:     Update $\mathcal{F}(\cdot)$ by **Algorithm 1** as the scenario. |
| | 12:    **end for** |
| | 13: **end for** |

$$\mathcal{F}_{\text{env}}(\tilde{s}) = \beta_{\text{P}} \mathcal{F}_{\text{PAN}}(\mathcal{P}_{\mathcal{A}}(\tilde{s}, \tilde{x})), \tag{13}$$

where $\beta_{\text{P}}$ represents the feature normalization coefficient. The training process of PAN is shown in Algorithm 2.

Unlike GNN and PAN networks, BPN, which is utilized for battery prediction in the agricultural application, is designed for intermediate decision-making during transitions of tasks. More specifically, we define a task-transition model $\mathcal{Q}_{EM}(\mathcal{S}_{\mathcal{B}}, \mathcal{A}_{\mathcal{B}})$, which determines whether the battery capacity runs short, so that the task needs to be switched from performing data collection service to returning to charging.

We use GNN-EAC or PAN-EAC to pre-train such a task-transition model. Then, we utilize such a model for decision-making as follows. First, we utilize $\mathcal{Q}_{EM}(\mathcal{S}_{\mathcal{B}}, \mathcal{A}_{\mathcal{B}})$ to generate the collection of task-transition states $\mathcal{S}_{EB}$ and the collection of corresponding battery labels $\mathcal{X}_{EB}$ by the interaction with the environment, where $\mathcal{S}_{EB}$ consists of task-transition state $s_E$. After that, we update the BPN layers by the loss function $\mathcal{J}_{\mathcal{BP}}(\epsilon)$ as follows:

$$\mathcal{J}_{\mathcal{BP}}(\epsilon) = ||\{S_{EB}, X_{EB}\} - \{S_{EB}, X'_{EB}\}||^2, \tag{14}$$

where $\epsilon$ represents the updating parameter and $\mathcal{X}'_{EB}$ represents the collection of predicted battery labels. Finally, we have a prediction model $\mathcal{F}_{\text{BPN}}(s_E)$ that determines whether the current state is the task-transition state or not. The training process of BPN is shown in Algorithm 3.

The overall training process of FM-EAC is shown in Algorithm 4. Note that for sequential tasks, $\mathcal{Q}(\cdot)$ is composed of two independent critic networks, together with their target networks. While for concurrent tasks, $\mathcal{Q}(\cdot)$ consists of the primary and secondary critic networks, together with their target networks. Furthermore, the design of GNN, PAN, and BPN enables efficient and robust environmental feature extraction. Thus, our proposed FM-EAC algorithm can not only interact with the environment but also transfer to other diverse environments. In practice, they can be replaced by other neural networks or feature matrices according to user-specific requirements.

## 4 PERFORMANCE EVALUATION

### 4.1 EXPERIMENTAL SETUP

We conducted experiments in two representative applications: an urban and an agricultural one, with formulations detailed in MDPs. All simulations were conducted on a MacBook Pro equipped with an Apple M4 chip (12-core CPU, 16-core GPU) and 24 GB of unified memory.

In the urban application, 4 UAVs were deployed within an $800\,\text{m} \times 800\,\text{m}$ area. The number of GDs ranged from 20 to 50, while the number of PDs varied from 0 to 50. The distribution of buildings was based on real-world *Digital Surface Model (DSM)*. The distribution of BSs and IoT devices was extracted from *OpenCellid* OpenCellid contributors (2025). The pedestrian traces were simulated

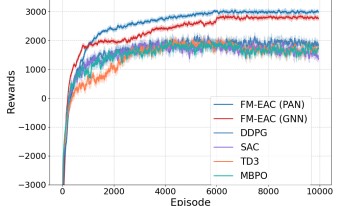
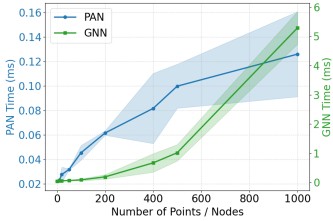

(a) The performance of training reward in urban application.

(b) The performance of training reward in agricultural application.

(c) Average inference time (ms) for PAN network and GNN.

Figure 2: The performance of training reward and time complexity.

Table 1: Comparison of average reward, computation time, and performance metrics in different scenarios.

| Algorithm | Reward ↑ | | Online Time ↓ (ms) | | Offline Time ↓ (ms) | | Urban | | Agriculture |
| | Urban | Agri. | Urban | Agri. | Urban | Agri. | QoS ↑ | Time ↓ (s) | AoI ↓ |
|---|---|---|---|---|---|---|---|---|---|
| ACO | 1352.41 (±190.75) | -60.76 (±0.28) | – | – | 55900 | 2720 | 5.81 | 99.2 | 2.65 |
| GA | 802.36 (±140.64) | 107.27 (±0.45) | – | – | 72960 | 2540 | 4.82 | 98.9 | 2.56 |
| PSO | 10.21 (±16.11) | 114.15 (±0.12) | – | – | 45880 | 2530 | 1.19 | 99.3 | 2.48 |
| DDPG | 362.83 (±72.96) | 1896.41 (±488.91) | 16.73 | 25.63 | 49.20 | 0.62 | 4.53 | 81.2 | 1.55 |
| TD3 | 410.05 (±62.06) | 1653.48 (±451.54) | **15.16** | 22.91 | 79.50 | **0.61** | 8.26 | 98.3 | 1.77 |
| SAC | 445.87 (±49.72) | 1543.85 (±435.97) | 17.34 | **13.54** | 74.30 | 1.05 | 7.47 | 96.7 | 1.80 |
| MBPO | 134.14 (±80.19) | 1635.42 (±461.81) | 44.85 | 37.76 | 73.50 | 0.75 | 7.29 | 98.2 | 1.97 |
| **PAN-EAC** | 1391.75 (±62.38) | **2402.47** (±5.42) | 16.96 | 37.00 | 69.50 | 0.68 | 8.00 | **76.3** | **1.10** |
| **GNN-EAC** | **1400.30** (±59.38) | 2153.84 (±7.50) | 35.94 | 74.96 | **36.30** | 5.02 | 8.08 | 77.1 | 1.27 |

by *SUMO randomTrips* Lopez et al. (2018). The communication model adopted was based on the 3GPP 36.873 standard 3GPP (2012), ensuring realistic urban channel characteristics.

In the agricultural application, 4 UAVs were deployed, operating within a $400 \text{ m} \times 400 \text{ m}$ area. A total of 400 WSs were uniformly distributed on the ground in this area. The terrain was synthetically generated based on realistic geographic features such as hills, plains, ravines, and valleys, simulating complex rural topography. Data transmission was carried out using a data transmission protocol with retransmission and verification mechanisms to ensure reliability.

## 4.2 COMPARATIVE STUDY

In the comparative study, we select the latest MFRL models, SAC Haarnoja et al. (2018) and TD3 Dankwa & Zheng (2020), as the base algorithms for FM-EAC in urban and agricultural applications, respectively. To enhance generalizability, we train the FM-EAC models on 3-5 out of 10 maps in different scenarios, and test them on a random map.

Meanwhile, we select the following state-of-the-art baselines. For the MBRL algorithms, MORel Kidambi et al. (2020) can only be used for offline tasks. COMBO Yu et al. (2021) and Dreamer require large-scale training samples and intensive parameter tuning. Thus, we select MBPO Janner et al. (2019) for comparison. For MFRL algorithms, DQN Mnih et al. (2013) is not feasible for continuous tasks. PPO Schulman et al. (2017) suffers from sample inefficiency due to its on-policy nature. Therefore, we select DDPG Lillicrap et al. (2016), SAC, and TD3 for comparison, all of which are off-policy algorithms following the actor-critic framework. We also compare FM-EAC with meta-heuristic methods, including ant colony optimization (ACO) Dorigo et al. (2006), particle swarm optimization (PSO) Kennedy & Eberhart (1995), and genetic algorithm (GA) Immanuel & Chakraborty (2019). Due to the inherent structure of the above algorithms, they can only be trained on a single map.

As presented in Figs. 2a, 2b, and Table 1, the proposed FM-EAC outperforms all baselines in average reward, convergence speed, and convergence stability. Furthermore, GNN-EAC exhibits slightly higher performance in the urban application, while PAN-EAC achieves slightly higher performance in the agricultural application.

Table 2: Average reward for the ablation study.

| Algorithm | Urban ↑ | Agriculture ↑ |
|---|---|---|
| PAN-OAC | 592.26 ($\pm$109.12) | — |
| GNN-OAC | 903.91 ($\pm$89.72) | — |
| NFM-EAC | 472.62 ($\pm$112.19) | 1623.45 ($\pm$9.91) |
| **PAN-EAC** | 1391.75 ($\pm$62.38) | **2402.47** ($\pm$5.42) |
| **GNN-EAC** | **1400.30** ($\pm$59.38) | 2153.84 ($\pm$7.50) |

The model-based MBPO has the lowest reward due to the lack of transferability. Similarly, existing MFRL methods lag behind FM-EAC since they can only be trained on a specific map, so their policies become less effective when adapting to new environments, especially in the highly dynamic urban application. Although ACO and GA achieve good performance in the urban application, they fall behind in the agricultural application, since they are not feasible for sequential tasks where the objective alters during the flight.

In contrast, the proposed FM-EAC utilizes feature models to extract environmental features from diverse maps, achieving the highest generalizability and transferability. In addition, by using an enhanced actor-critic framework, the policies of different tasks can be jointly and smoothly updated, yielding the highest convergence value and stability.

### 4.3 ABLATION STUDY

In the ablation study, we sequentially remove components of GNN-EAC and PAN-EAC to evaluate their individual contributions. First, we replace the enhanced actor-critic framework with the original actor-critic (OAC) structure, resulting in GNN-OAC and PAN-OAC. Notably, the structures of EAC and OAC are identical in the agricultural application, thus they are interchangeable. Table 2 shows that GNN-EAC and PAN-EAC are superior to GNN-OAC and PAN-OAC due to the decoupling of actors and critics for different tasks. Next, we remove the feature extraction models entirely, yielding NFM-EAC (non-feature model enhanced actor-critic). The absence of feature models leads to significant drops in average reward, especially in the highly dynamic urban application.

### 4.4 INFERENCE TIME STUDY

As shown in Table 1, we compare the inference time (in milliseconds) of FM-EAC and the baselines across two types of tasks: offline and online tasks, depending on whether the policies are trained before or during execution. Additionally, Fig. 2c depicts how the inference times of PAN and GNN increase with the number of points/nodes (e.g., UAVs, IoT devices, and WSs). It can be observed that the GNN network has a steeper growth slope than the PAN network; therefore, it has a higher time complexity.

ACO, GA, and PSO, with their iterative and multi-sample nature, do not support incremental updates and fast responses for online tasks. They operate using a population of candidate solutions, and each member of the population requires a separate evaluation per iteration. Besides, they rely on randomized decision-making, so they are less sample efficient and have significantly higher offline inference time.

For online tasks, TD3 and SAC show the shortest online inference time in urban and agricultural applications, respectively, while the remaining MFRL and PAN-EAC algorithms show comparable and slightly higher online inference times. This is thanks to the lightweight design of PAN and actor-critic networks. Unlike them, MBPO and GNN-EAC encompass more computationally intensive models, so their online inference times are much longer.

For offline tasks, in the urban application where the number of IoT devices $\leq 100$, GNN-EAC exhibits the shortest offline inference time, since the GNN network can rapidly extract rich environmental features (e.g., adjacency matrices among UAVs and IoT). However, in the agricultural application, GNN-EAC has the longest offline inference time because the massive number of WSs (i.e., 400) leads to a heavy computational burden. Meanwhile, the offline inference times of MBRL, MFRL, and PAN-EAC are similar.

In summary, the online inference time of PAN-EAC is lower than that of GNN-EAC. When the number of devices is small, the offline inference time of GNN-EAC is lower; when the number of devices is large, the offline inference time of PAN-EAC is lower. In practice, we can choose between these alternative networks or customize the feature extraction model according to user-specific requirements.

### 4.5 REASONING STUDY

The reasoning study evaluates performance metrics compared with baselines. The metrics in the urban application include the average QoS of IoT devices (utility values with imaginary units) and average task completion time of UAVs (in seconds). Table 1 shows that TD3 archives the highest QoS. PAN-EAC and GNN-EAC, despite with 3.15% and 2.18% lower AoI, reduce task completion time by 22.38% and 21.57%, respectively. Therefore, we can conclude that FM-EAC can better draw the trade-off between QoS and task completion time. The performance metric in the agricultural application is the average AoI of the WSs (utility values with imaginary units). PAN-EAC and GNN-EAC achieve the lowest AoI, showing their superior performance.

## 5 RELATED WORKS

Most existing research on generalized reinforcement learning algorithms has centered on meta-reinforcement learning, which trains a meta-policy on various tasks to embed prior knowledge and facilitate fast adaptation. Some research utilized recurrent neural networks (RNNs) to embed an agent's learning process Duan et al. (2016); Agarwal et al. (2024). Others used gradient descent for policy adaptation in the inner loop Finn et al. (2017); Rakelly et al. (2019). However, the parameter tuning for the above methods requires intensive efforts, and the above methods increase generalizability at the cost of degrading performance. Inspired by few-shot learning in supervised tasks, feature metrics or external memory have also been used for policy adaptation. Our approach inherits such an idea, using a feature-based model to extract the feature representations that can rapidly adapt to various environments.

Meanwhile, most existing research on multi-task control concentrated on discrete-continuous hybrid action spaces. P-DQN Xiong et al. (2018) is a parametrized DQN framework for the hybrid action space where discrete actions share the continuous parameters. Similarly, HD3 Jiang & Ji (2019), a distributed dueling DQN algorithm, was proposed to produce joint decisions by using three sequences of fully connected layers. Hybrid SAC Delalleau et al. (2019) is an extension of the SAC algorithm, where an actor computes a shared hidden state representation to produce both the discrete and continuous distributions. HPPO Fan et al. (2019), a hybrid architecture of actor-critic algorithms, was proposed to decompose the action spaces along with a critic network to guide the training of all sub-actor networks. However, the above works considered an identical objective for all tasks, neglecting the interrelationship and heterogeneity between them. In contrast, FM-EAC can effectively handle interrelated tasks with their respective goals through the enhanced actor-critic framework.

## 6 CONCLUSION

We propose a generalized model, FM-EAC, that integrates planning, acting, and learning for multi-task control in dynamic environments, and leverages the strengths of MBRL and MFRL. Our feature model improves the generalizability and transferability across scenarios, and our enhanced actor-critic framework supports simultaneous policy updates, promoting efficient and effective learning across diverse objectives. The performance of FM-EAC is validated through urban and agricultural applications. Experimental results demonstrate that FM-EAC consistently outperforms state-of-the-art MBRL and MFRL algorithms. Moreover, GNN-based and PAN-based FM-EAC achieve comparable performance, while exhibiting distinct time efficiency for online and offline tasks. In the future, we will extend FM-EAC to a wider range of environments and practical domains, such as multi-robot control, multi-user autonomous driving, and multi-player games.

REPRODUCIBILITY STATEMENT

This work adheres to the standards of reproducibility. The link to the demonstration code of the proposed algorithms together with the explanation is provided in Appendix A.2. Additionally, scenario description, problem formulation, and parameter settings regarding the exemplary urban and agricultural applications are presented in Appendix A.3, A.4, and A.5, respectively. For anonymity, information regarding city, terrain, and document paths has been omitted. Following acceptance, we will release the full executable code for function validation.

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

# A  APPENDIX

## A.1  THE USE OF LARGE LANGUAGE MODELS (LLMS)

We declare that LLMs are only used for the retrieval and discovery of related works. They are **NOT** used for polishing writing, research ideation, problem formulation, algorithm development, result generation, or other purposes.

## A.2  EXPLANATION OF DEMONSTRATION CODE

The demonstration code can be downloaded from link to demonstration code. In the demonstration code, there are six files. "ReadMe.md" explains the code in general. "Folder_structure" shows the architecture of the entire code, which will be released upon acceptance. The other four files are the demonstration code detailed below.

### A.2.1 General Description

For anonymity, information regarding city, terrain, and document paths has been omitted. To facilitate understanding of the proposed algorithms, only agent-related code segments are presented here. The complete folder structure is documented in the accompanying "Folder_structure.pdf". Following acceptance, we will release the full executable code for function validation.

The demonstration code includes the following four files.

*agri_eac_gnn_model.py*: Implements network architectures for GNN, BPN, and FM-EAC tailored to agricultural applications.

*agri_eac_pan_model.py*: Implements network architectures for PAN, BPN, and FM-EAC tailored to agricultural applications.

*urban_eac_gnn_model.py*: Implements network architectures for GNN and FM-EAC tailored to urban applications.

*urban_eac_pan_model.py*: Implements network architectures for PAN and FM-EAC tailored to urban applications.

### A.2.2 Detailed Description of Key Modules

The explanation of key modules (i.e., elements and functions) in each file is detailed as follows.

*agri_eac_gnn_model.py*:

- Actor: Policy network generating actions conditioned on input states.
- Critic: Evaluation network estimating Q-values for state-action pairs.
- BatteryPredictionNetwork: Predicts energy consumption based on state and environmental features.
- normalize_adjacency_matrix(A): Normalizes adjacency matrices used in graph convolution layers.
- GCNLayer: Single graph convolutional layer processing node features with normalized adjacency.
- GNN: Two-layer graph neural network producing a global graph representation via node features and adjacency.
- GNN_Agent:
- compute_gnn_loss(batch_state, batch_action, batch_reward, batch_done): Computes critic loss for training GNN and critic networks.
- choose_action(state, explore=True): Selects an action given the current state, optionally with exploration noise.
- store_transition(state, action, reward, next_state, done): Saves experience tuples into replay buffer, managing buffer capacity.
- update(): Samples batches from replay buffer and updates GNN, Critic, and Actor networks with soft target updates.
- save(i, path, eps): Saves model weights (Actor, Critics, GNN) to checkpoint files.
- load(i, path, eps): Loads model weights from checkpoint files; raises error if files are absent.

*agri_eac_pan_model.py*:

- Actor: Policy network generating actions based on input states.
- Critic: Value network estimating Q-values for state-action pairs.
- BatteryPredictionNetwork: Estimates energy consumption from input state and environment features.
- PointArrayFeatureExtractor: Extracts features from input environmental point array data.

- PAN_Agent:

- choose_action(state, explore=True): Selects action given state, optionally applying exploration noise.

- store_transition(state, action, reward, next_state, done): Stores experience tuples in replay buffer, handling buffer size constraints.

- update(): Samples from replay buffer and updates Critic and Actor networks, including soft target network updates.

- save(i, path, eps): Persists model parameters (Actor, Critics) to disk with iteration and episode identifiers.

- load(i, path, eps): Loads model parameters from saved checkpoints; raises error if missing.

*urban_eac_gnn_model.py*:

- Actor: Policy network producing two types of actions from input states:

- Primary action (mean mu and standard deviation std): continuous actions modeled as a Gaussian distribution.

- Secondary action (softmax_out): categorical distribution over three discrete options via softmax.

- Critic_Pri: Value network estimating Q-values for state-primary action pairs.

- Critic_Sec: Value network estimating Q-values for state-secondary action pairs.

- normalize_adjacency_matrix(A): Adds self-loops and normalizes adjacency matrix for graph convolutional layers.

- GCNLayer: Single graph convolution layer performing adjacency normalization and learnable feature transformation.

- GNN: Two-layer graph neural network processing node features and adjacency, outputting global graph features by mean pooling.

- GNN_Agent:

- compute_gnn_loss(batch_state, batch_action): Computes loss on critic Q-values to update the GNN network by encouraging higher Q-values.

- update(): Conducts a training step by sampling batches, computing target Q-values, updating critics and actors, and applying GNN loss optimization.

- choose_action(state): Samples actions combining continuous primary actions and discrete secondary actions from the actor output.

- save(i, path, eps): Saves model weights (actor, critics, GNN) to checkpoint files.

- load(i, path, eps): Loads model weights from checkpoint files.

*urban_eac_pan_model.py*:

- Actor: Policy network outputting two action types from the input state:

- Primary action (mean mu and standard deviation std): continuous actions modeled as a Gaussian distribution, dimension (action_dim - 3).

- Secondary action (softmax_out): categorical distribution over three discrete options via softmax.

- Critic_Pri: Value network estimating Q-values for state and primary action pairs (continuous).

- Critic_Sec: Value network estimating Q-values for state and secondary action pairs (discrete, one-hot encoded).

- PointArrayFeatureExtractor: Extracts features from environmental point array inputs.

- PAN_Agent:

- update(): Executes a training iteration by sampling from replay buffer, computing target Q-values, optimizing critics via MSE loss, and updating actor networks to maximize expected Q-values.

- choose_action(state): Samples combined continuous and discrete actions from actor outputs.

- save(i, path, eps): Saves current model parameters (actor and critics) with iteration and episode labels.

- load(i, path, eps): Loads model parameters from saved checkpoints.

### A.3 SCENARIO DESCRIPTION

This section illustrates the exemplary scenarios, including urban and agricultural applications. Additionally, the system model for each application is presented.

#### A.3.1 URBAN APPLICATION OVERVIEW

In the urban application, we consider multiple metropolitan areas where a collaborative system composed of $n_{\text{UAV}}$ unmanned aerial vehicles (UAVs) is deployed to simultaneously perform material delivery and mobile edge computing (MEC) services. The set of UAVs is denoted by UAV = $\{\text{UAV}_1, \ldots, \text{UAV}_{n_{\text{UAV}}}\}$.

The position of $\text{UAV}_i$ at time $t$ is represented by the 3D vector $\boldsymbol{Pu}(i,t) = \{Pu_x(i,t), Pu_y(i,t), Pu_z(i,t)\}$, where $Pu_x(i,t)$, $Pu_y(i,t)$, and $Pu_z(i,t)$ denote the spatial coordinates along the $x$-, $y$-, and $z$-axes, respectively.

There are two services in the urban application: the package delivery service for the UAVs from the origin to the destination, and the MEC service from the UAVs to the IoT devices. There are two concurrent tasks accordingly. The primary (PRI) task is the joint trajectory planning for both services, and the secondary (SEC) task is the task offloading decision for the MEC service.

Each UAV is required to carry out delivery tasks within designated regions while adhering to several constraints: avoiding collisions with buildings, maintaining sufficient signal quality from base stations (BSs), and satisfying the quality-of-service (QoS) requirements of ground IoT devices. The cruising altitude of each UAV must remain within a bounded range, i.e., $Pu_z(i,t) \in [Pu_{z\text{-min}}, Pu_{z\text{-max}}]$.

For each $\text{UAV}_i$, the delivery task starts from an initial position $\boldsymbol{Pu}_{\text{start}}(i) = \boldsymbol{Pu}(i,0)$ and proceeds toward a target destination $\boldsymbol{Pu}_{\text{end}}(i) = \boldsymbol{Pu}(i, T_{\text{end}})$, where $T_{\text{end}}$ denotes the task deadline. The 3D operational task space is defined as: $\mathbb{R}^3 : \{x \in [Pu_{x-\text{min}}, Pu_{x-\text{max}}], y \in [Pu_{y-\text{min}}, Pu_{y-\text{max}}], z \in [Pu_{\{z-\text{min}}, Pu_{z-\text{max}}]\}$. Due to limited onboard battery, each UAV must complete its task within $T_{\text{end}}$ and be within a predefined threshold distance $d_{\text{end}}$ of its destination to be considered successful.

In parallel with delivery, UAVs provide MEC services to IoT subscribers on the ground, including pedestrian-carrying devices (PDs) and ground devices (GDs), to reduce computation latency. The GDs are stationary while the PDs are moving with the pedestrians. Each UAV is equipped with an omnidirectional antenna of fixed gain and can simultaneously maintain uplink connections with up to $m$ IoT devices due to limited channel capacity. To ensure QoS, UAVs must dynamically allocate their available power resources among the connected devices.

Let $n_{\text{IoT}}$ denote the total number of IoT devices, represented as IoT = $\{\text{IoT}_1, \ldots, \text{IoT}_{n_{\text{IoT}}}\}$. These devices, sharing the same communication frequency, are the main targets of MEC services. Due to the mobility of PDs, the position of device $\text{IoT}_i$ at time $t$ is defined as $\boldsymbol{Pi}(i,t) = \{Pi_x(i,t), Pi_y(i,t), Pi_z(i,t)\}$, where $Pi_z(i,t)$ reflects the ground elevation, and $Pi_x(i,t), Pi_y(i,t)$ represent the planar location.

Each IoT device establishes an IoT-UAV communication link to receive downlink signals and upload data. As the channel capacity improves, the corresponding MEC QoS also increases. The computational tasks generated by IoT devices are both heterogeneous and dynamic: different devices request different service types, which may change over time based on certain probability distributions.

In addition to UAVs and IoT devices, we also consider $n_{\text{BS}}$ base stations (BSs), denoted as BS = $\{\text{BS}_1, \ldots, \text{BS}_{n_{\text{BS}}}\}$, deployed atop buildings to provide 5G communication services to UAVs.

Each BS is equipped with three unidirectional antennas, spaced $120°$ apart, forming three distinct coverage sectors. The set of all antennas is represented as $\text{ANT} = \{\text{ANT}_1, \ldots, \text{ANT}_{3n_{\text{BS}}}\}$.

Depending on urban obstacles, communication channels are categorized as line-of-sight (LoS) or non-line-of-sight (NLoS) links. For each UAV, the BS with the strongest received signal is selected as its serving BS, while signals from other BSs are treated as interference.

### A.3.2 URBAN APPLICATION MODEL DESIGN

In the System Model, we considered the energy model, antenna model, and communication model.

**Energy Consumption:**

Each UAV performs material delivery and MEC services, including task offloading and computation. The cumulative energy consumption of $\text{UAV}_i$ up to time $t$ is:

$$EC(i,t) = EC_{\text{cmp}}(i,t) + EC_{\text{comm}}(i,t) + EC_{\text{fly}}(i,t), \tag{15}$$

where the components represent energy consumption for computation, communication, and flight, respectively.

The cumulative computation energy is:

$$EC_{\text{cmp}}(i,t) = \int_0^t Pw_{\text{cmp}} \, dt, \tag{16}$$

with constant computation power $Pw_{\text{cmp}}$.

Communication energy is:

$$EC_{\text{comm}}(i,t) = \int_0^t (Pw_{ut} + Pw_{ur}) \, dt, \tag{17}$$

where $Pw_{ut}$ and $Pw_{ur}$ are transmission and reception powers, respectively.

Flight energy is:

$$EC_{\text{fly}}(i,t) = \int_0^t Pw_{\text{fly}}(i,t) \, dt, \tag{18}$$

with $Pw_{\text{fly}}(i,t)$ defined as:

$$Pw_{fly}(i,t) = \begin{cases} \frac{m_{\text{UAV}} g^{3/2}}{\sqrt{2\rho_{\text{air}} A_{\text{UAV}} \eta}}, & |\boldsymbol{v}(i,t)| < v_{\text{th}}, \\ c_1 |\boldsymbol{v}(i,t)|^2 + c_2 \frac{v_x^2(i,t) + v_y^2(i,t)}{|\boldsymbol{v}(i,t)|^3} + mg|\boldsymbol{v}(i,t)|, & |\boldsymbol{v}(i,t)| \geq v_{\text{th}}, \end{cases} \tag{19}$$

The aerodynamic parameters are:

$$c_1 = \frac{1}{2}\rho A_{\text{UAV}} C_d, \tag{20}$$

$$c_2 = \frac{m_{\text{UAV}}^2}{\eta \rho_{\text{air}} n_{\text{prp}} \pi R_{\text{prp}}^2}, \tag{21}$$

$$A_{\text{UAV}} = A_{\text{surf}} + n_{\text{prp}} \pi R_{\text{prp}}^2. \tag{22}$$

The remaining battery level is:

$$BR(i,t) = BC - EC(i,t), \tag{23}$$

where $BC$ is the battery capacity.

**Antenna:**

Each BS has three directional antennas separated by $120°$, each serving one sector. Each antenna consists of an $m_{\text{ULA}} \times n_{\text{ULA}}$ ULA, with element spacing $d_{\text{ULA}}$.

Based on 3GPP **?**, the total attenuation is:

$$A_T(\theta, \phi) = A_H(\theta) + A_V(\phi), \tag{24}$$

with horizontal and vertical attenuations:

$$A_H(\theta) = -\min\left\{12\frac{\theta - \theta_{\text{main}}}{\Theta_{\text{3dB}}}, 30\text{dB}\right\}, \tag{25}$$

$$A_V(\phi) = -\min\left\{12\frac{\phi - \phi_{\text{main}}}{\Phi_{\text{3dB}}}, 30\text{dB}\right\}. \tag{26}$$

The beamformed array factor is:

$$AF(\theta, \phi) = |\sum_{m=0}^{m_{\text{ULA}}-1}\sum_{n=0}^{n_{\text{ULA}}-1} \exp(jk_w d_{\text{ULA}}(m\sin(\theta)\cos(\phi)) + n\sin(\theta)\sin(\phi)|^2, \tag{27}$$

where $k_w = \frac{2\pi f_{\text{BS}}}{c}$.

Antenna gain:

$$G(\theta, \phi) = G_{\text{max}} + A_T(\theta, \phi) + 10\log_{10}(AF(\theta, \phi)), \tag{28}$$

$$G_{\text{max}} = G_{\text{element}} + 10\log_{10}(m_{\text{ULA}} \times n_{\text{ULA}}). \tag{29}$$

**Communication:**

To account for urban LoS and NLoS cases:

$$PL(\text{Tra}, \text{Rec}) = \begin{cases} 28.0 + 22\log_{10}(d_{\text{TS}}) + 20\log_{10}(f_c), & LoS, \\ \\ -17.5 + 20\log_{10}(\frac{40\pi f_c}{3}) \\ \\ +[46 - 71\log_{10}(H_{\text{TS}})]\log_{10}(d_{\text{TS}}), & NLoS, \end{cases} \tag{30}$$

Signal power from antenna $\text{ANT}_j$ to $\text{UAV}_i$:

$$Pw_{\text{U-B}}(i, j, t) = Pw_{ur} + Pw_{bt} + G(\theta_{i,j}(t), \phi_{i,j}(t)) - PL(Pu(i,t), \text{ANT}_j), \tag{31}$$

The SINR:

$$\text{SINR}_{\text{U-B}}(i, t) = \frac{\mathcal{S}_{\text{U-B}}(i, t)}{\mathcal{I}_{\text{U-B}}(i, t) + P_n}, \tag{32}$$

$$\mathcal{S}_{\text{U-B}}(i, t) = \max_j Pw_{\text{U-B}}(i, j, t), \tag{33}$$

$$\mathcal{I}_{\text{U-B}}(i, t) = \sum_j Pw_{\text{U-B}}(i, j, t) - \mathcal{S}_{\text{U-B}}(i, t), \tag{34}$$

$$P_n = k_B T_K Bw. \tag{35}$$

Each IoT device uses an omnidirectional antenna with gain:

$$Pw_{\text{U-I}}(i, j, t) = Pw_{ur} \cdot \delta_{rj} + Pw_{it} + G_{\text{IoT}} - PL(\boldsymbol{Pu}(i,t), Pi(j,t)), \tag{36}$$

Each UAV connects to at most $m$ IoT devices:

$$\{j_1, \ldots, j_m\} = \arg\max_j (Pw_{\text{U-I}}(i, j, t)), \tag{37}$$

with power allocation:

$$\boldsymbol{\delta}_r(i, k) = \{\delta_{rj_1}, \ldots, \delta_{rj_m}\}, \sum\boldsymbol{\delta}_r \leq \epsilon. \tag{38}$$

The SINR and capacity for link $(i, j_k)$:

$$\text{SINR}_{\text{U-I}}(i, j_k, t) = \frac{Pw_{\text{U-I}}(i, j_k, t)}{\mathcal{I}_{\text{U-I}}(i, t) + P_n}, \tag{39}$$

$$\mathcal{I}_{\text{U-I}}(i, t) = \sum_j Pw_{\text{U-I}}(i, j, t) - \sum_{j_k} Pw_{\text{U-I}}(i, j_k, t), \tag{40}$$

$$\mathcal{C}(i, j_k, t) = Bw \cdot \log_2(1 + \text{SINR}_{\text{U-I}}(i, j_k, t)). \tag{41}$$

### A.3.3 AGRICULTURAL APPLICATION OVERVIEW

In the agricultural application, we consider multiple hilly farmlands with diverse terrains, where a collaborative system composed of $n_{\text{UAV}}$ UAVs is deployed to perform data collection servcie. The UAV set is denoted by $\text{UAV} = \{\text{UAV}_1, \ldots, \text{UAV}_{n_{\text{UAV}}}\}$. Each $\text{UAV}_i$ departs from its designated docking station (DS) $\text{DS}_i$ and must return to $\text{DS}_i$ for recharging before its battery is depleted. When $\text{UAV}_i$ enters the charging zone, another fully charged UAV is dispatched from the same DS to seamlessly continue $\text{UAV}_i$'s mission. The set of DS is denoted by $\text{DS} = \{\text{DS}_1, \ldots, \text{DS}_{n_{\text{UAV}}}\}$.

To accurately capture environmental variations in crop growth, a wireless sensor network (WSN) consisting of $n_{\text{WS}}$ wireless sensors (WSs), represented as $\text{WS} = \{\text{WS}_1, \ldots, \text{WS}_{n'}\}$, is deployed to monitor soil and atmospheric conditions. All WSs are assumed to be equipped with identical low-power modules for both sensing and wireless communication. These sensors continuously record environmental parameters such as humidity, temperature, and soil pH, and periodically transmit data to nearby UAVs. WSs of different types may have distinct sensing and broadcasting cycles. UAVs receive this data and relay it to DSs, thereby supporting situational awareness for agricultural monitoring and early warning systems.

In the agricultural application, each UAV performs two sequential tasks: the data collection (COL) task, where data is gathered from ground WSs, and the return-to-home (RTH) task, where the UAV travels to the DS for recharging. During the COL phase, UAVs focus on exploration, traversing the farmland to continuously reduce the AoI of the WSs through timely data collection. When a UAV estimates that its residual energy becomes short based on the battery prediction network (BPN), it transitions to the RTH mode. In this phase, the UAV prioritizes a safe return to the DS while opportunistically collecting data along the route, thereby balancing the trade-off between mission continuity and energy sufficiency.

Similar to the urban application described earlier, the position of $\text{UAV}_i$ at time $t$ is denoted by $\boldsymbol{Pu}(i,t) = \{Pu_x(i,t), Pu_y(i,t), Pu_z(i,t)\}$. Each UAV performs data collection across the farmland while satisfying several constraints: avoiding collisions with other UAVs, maintaining proper flight altitudes, minimizing the overall Age of Information (AoI) of the WSs, and returning to the corresponding DSs before battery exhaustion.

For each $\text{UAV}_i$, the 3D task space can be defined as $\mathbb{R}^3 : \{x \in [Pu_{x-\min}, Pu_{x-\max}], y \in [Pu_{y-\min}, Pu_{y-\max}], z \in [Pu_{\{z-\min}, Pu_{z-\max}]\}$. Given the UAVs' limited battery capacities, each mission must be completed within a time horizon $T_{\text{end}}$, and the UAV must reach within a predefined threshold distance $d_{\text{end}}$ of its DS for the return operation to be deemed successful.

### A.3.4 AGRICULTURAL APPLICATION MODEL DESIGN

In the System Model, we considered the energy model and communication model.

**Energy Consumption:**

The energy model is similar to the urban application.

**Communication:**

The AoI of each $\text{WS}_j$ in WSN in each time slot $t$ can be represented as $\text{AoI}_j(t)$. The AoI of WSs increases during each updating interval $T_{\text{Update}_{\text{AoI}}}$. WSs of different types have different $T_{\text{Update}_{\text{AoI}}}$ but the same AoI threshold $\text{AoI}_{\max}$. Each $\text{WS}_j$ broadcasts connection requests to surrounding UAVs per $T_{\text{Update}_{\text{AoI}}}$. Afterwards, $\text{WS}_j$ transmits data packets to its connected $\text{UAV}_i$. If the transmission is successful, $\text{UAV}_i$ will send a message to $\text{WS}_j$ to reset $\text{AoI}_j(t)$ and disconnect. Otherwise, transmission timeout and $\text{WS}_j$ will prepare for re-transmission next time. Finally, $\text{UAV}_i$ sends the collected data packages to $\text{DS}_i$.

Then, we define the packet loss rate $\text{PLR}(P_i(t))$ in data transmission as:

$$\text{PLR}(P_i(t)) = 1 - (1 - \text{BER}(P_i(t))^L, \tag{42}$$

where $L$ represents the package length and $\text{BER}(P_i(t))$ represents the bit error rate at position $P_i(t)$. The binary phase shift keying data (BPSK) encoding method is adapted in this paper, so the $\text{BER}(P_i(t))$ can be represented as:

$$\text{BER}(P_i(t)) = Q\left(\sqrt{2 \cdot \text{SINR}(P_i(t))}\right), \tag{43}$$

where $\text{SINR}(P_i(t))$ represents the signal-to-interference-plus-noise ratio (SINR) at position $P_i(t)$, and $Q(x)$ means Q-function in communication theory. They can be represented as:

$$\text{SINR}\,(P_i(t)) = \frac{\sum_{k,\text{WS}_k \in \text{WS}}^{n} Pw_{wt}(t) + G_{\text{WS}}(t) - PL(\text{UAV}_i, \text{WS}_j)(t)}{\sum_{k,\text{WS}_k \notin \text{WS}}^{n} Pw_{wt}(t) + G_{\text{WS}}(t) - PL(\text{UAV}_i, \text{WS}_j)(t)}, \tag{44}$$

$$Q(x) = \frac{1}{\sqrt{2\pi}} \int_x^\infty e^{-\frac{t^2}{2}} dt \approx \frac{1}{2} e^{-\frac{x^2}{2}}, \tag{45}$$

where $Pw_{wt}(t)$ represents the WS transmission power, $G_{\text{WS}}$ represents the gain of WSs, and $PL(\text{UAV}_i, \text{WS}_j)(t)$ represents the path loss between $\text{UAV}_i$ and $\text{WS}_j$.

### A.4 Markov Decision Process Formulation

#### A.4.1 MDP for Urban Application

In the urban application, the UAVs perform two tasks simultaneously with a priority order. We assume that the primary (PRI) task is the joint trajectory planning of UAVs for package delivery and MEC service, and the secondary (SEC) task is the computation offloading from IoT devices to UAVs. Therefore, we formulate the MDP as follows:

- **Observations**: The observation space provides information on UAVs, including their current positions, destinations, and the distances between them. Environmental information, such as the signal-to-interference-plus-noise ratio (SINR) values from BSs to UAVs and the locations and path loss of IoT devices, is also included. In FM-EAC, scenario-associated features equation 11 and equation 13 are fused into the observation space.
- **Actions**: The action of the PRI task is the flying speed vector for each UAV, and the action of the SEC task is the computational offloading rate for each PD and GD.
- **Rewards**: The PRI task reward aims to minimize the task completion time, subject to SINR, QoS, energy consumption, trajectory length, height variation, regional boundary, and collision constraints. The SEC task reward aims to maximize the QoS of IoT devices at each timestep. Notably, SEC could be regarded as a partial reward of PRI.

#### A.4.2 MDP for Agricultural Application

In the agricultural application, the UAVs perform two tasks sequentially depending on the predicted battery capacity of UAVs. When the predicted battery capacity is sufficient, the task is the data collection from WSs; when the predicted battery runs short, the task is returning to the DS while collecting data along the way. We call the former as collection (COL) task and the latter as return-to-home (RTH) task. Therefore, we formulate the MDP as follows:

- **Observations**: The observation space provides information on UAVs, including their current positions and nearby WSs' age of information (AoI) within the communication range. The observations also include the position of the closest UAV and DSs in the tasks COL and RTH, respectively. In FM-EAC, scenario-associated features equation 11 and equation 13 are fused into the observation space.
- **Actions**: The action of both COL and RTH tasks is the flying speed vector for each UAV.
- **Rewards**: The COL task reward aims to minimize the average AoI of WSs while flying the UAVs as low as possible w.r.t. terrain altitude to save energy. The RTH task reward aims to minimize the time to reach the DS for charging while minimizing the average AoI along the way.

### A.5 Parameter Settings in the Simulation

This section includes four tables: Table 3 and Table 4 present the environmental parameters and hyperparameters for the urban application, respectively. Table 5 and Table 6 present the environmental parameters and hyperparameters for the agricultural application, respectively. The environmental parameters are set according to real-world characteristics. The weight parameters are set so that each constraint term is comparable with each other. The hyperparameters are tuned to achieve the best training performance.

Table 3: Environmental Parameters for the Urban Application.

| Symbol | Definition | Value | Unit | Symbol | Definition | Value | Unit |
|---|---|---|---|---|---|---|---|
| $pu_{x-\min}$ | Task Space Left Edge | 0 | m | $pu_{x-\max}$ | Task Space Right Edge | 800 | m |
| $pu_{y-\min}$ | Task Space Back Edge | 0 | m | $pu_{y-\max}$ | Task Space Front Edge | 800 | m |
| $pu_{z-\min}$ | Task Space Bottom Edge | 180 | m | $pu_{z-\max}$ | Task Space Top Edge | 220 | m |
| $d_{\text{end}}$ | Advance End Task Distance | 50 | m | $V_{x-\max}$ | $x$ Dire. Maximum Velocity | 8 | m/s |
| $V_{y-\max}$ | $y$ Dire. Maximum Velocity | 8 | m/s | $V_{z-\max}$ | $z$ Dire. Maximum Velocity | 8 | m/s |
| $T_{\text{end}}$ | Maximum Mission Time | 100 | s | $n_{\text{UAV}}$ | UAV Number | $2-6$ | - |
| $n_{\text{BS}}$ | BS Number | 3, 4 | - | $n_{\text{IoT}}$ | IoT Device Number | $[0, 100]$ | - |
| $\epsilon$ | Allocation Proportion | 0.8 | - | $BC$ | Battery Capacity | 155520 | J |
| $Pw_{\text{cmp}}$ | Computation Power | 20 | W | $pw_{\text{ut}}$ | UAV Transmission Power | 20 | dBm |
| $Pw_{ur}$ | UAV Received Power | 20 | dBm | $m_{\text{UAV}}$ | UAV Mass | 0.2 | kg |
| $g$ | Gravitational Acceleration | 9.8 | - | $\rho_{\text{air}}$ | Air Density | 1.225 | kg/m$^3$ |
| $v_{\text{th}}$ | Hovering Speed Threshold | 0.1 | m/s | $C_d$ | Viscosity Coefficient | 0.5 | — |
| $n_{\text{prp}}$ | Propeller Number | 4 | - | $R_{\text{prp}}$ | Propeller Radius | 0.1 | m |
| $\eta$ | Mechanical Efficiency | 0.8 | - | $A_{\text{surf}}$ | UAV Fuselage Area | 0.01 | m$^2$ |
| $m_{\text{ULA}}$ | ULA Horizontal Dimension | 8 | - | $n_{\text{ULA}}$ | ULA Vertical Dimension | 8 | - |
| $d_{\text{ULA}}$ | Element Distance | 0.05 | m | $\theta_{\text{main}}$ | Horizontal Main Lobe Dire. | 0 | ° |
| $\phi\text{main}$ | Vertical Main Lobe Dire. | 80 | ° | $c$ | Light Speed | 3e8 | m/s |
| $\Theta_{\text{3dB}}$ | Horizontal 3dB Beam-width | 65 | ° | $\Phi_{\text{3dB}}$ | Vertical 3dB Beam-width | 65 | ° |
| $G_{\text{element}}$ | Antenna Element Gain | 5 | dB | $k_B$ | Boltzmann Constant | 1.38e-23 | - |
| $T_K$ | Temperature in Kelvin | 298 | K | $Bw$ | Bandwidth | 20 | MHz |
| $f_{BS}$ | BS Frequency | 3.5 | GHz | $f_{IoT}$ | IoT Device Frequency | 5.9 | GHz |
| $m$ | Up-link Limitation | 3 | - | $k_{\text{end}}$ | Discretized End Time | 100 | - |

Table 4: Hyperparameters for the Urban Application.

| Symbol | Definition | Value | Symbol | Definition | Value |
|---|---|---|---|---|---|
| $\alpha_1$ | Weight Parameter for Length | 1 | $\alpha_2$ | Weight Parameter for Flight Height | 0.75 |
| $\alpha_3$ | Weight Parameter for SINR | 2.5 | $\alpha_4$ | Weight Parameter for Energy Consumption | 0.1 |
| $\alpha_5$ | Weight Parameter for QoS | 0.75 | $\alpha_6$ | Weight Parameter for Out and Collision | 10 |
| $\alpha_7$ | Weight parameter for safety risk | 0.1 | $\alpha_8$ | Partial Reward Weight | 10 |
| $\gamma$ | Discounted Factor | 0.99 | $\mathcal{HN}$ | Normal Hidden Layers | 3 |
| $r_a$ | Learning Rate for Actor | $10^{-5}$ | $r_{ct}$ | Learning Rate for Total Reward Critic | $10^{-4}$ |
| $r_{cp}$ | Learning Rate for Partial Critic | $10^{-5}$ | $\mathcal{B}_{\mathcal{P}}$ | PAN Training Batch Size | 512 |
| $\mathcal{D}$ | Replay Buffer Size | $2^{16}$ | $\mathcal{B}$ | RL Training Batch Size | 256 |
| $epi_{\max}$ | Maximum Training Episode | 1000 | $epo_{\max}$ | PAN Pre-training Maximum Epoch | 100 |
| $\xi$ | Soft Update parameter | 0.01 | $k_{\max}$ | Maximum Timestep | 100 |
| $r_G$ | Learning Rate for GNN | $10^{-3}$ | $r_P$ | Learning Rate for PAN | $10^{-4}$ |
| $\tilde{S}$ | Scenario Number | 3 | $\tilde{X}$ | PMP Trace Number | 30 |
| $\beta_P$ | PAN Normalization Coefficient | 0.01 | $\beta_G$ | GNN Normalization Coefficient | 0.01 |

Table 5: Environmental Parameters for the Agricultural Application.

| Symbol | Definition | Value | Unit | Symbol | Definition | Value | Unit |
|---|---|---|---|---|---|---|---|
| $pu_{x-\min}$ | Task Space Left Edge | 0 | m | $pu_{x-\max}$ | Task Space Right Edge | 400 | m |
| $pu_{y-\min}$ | Task Space Back Edge | 0 | m | $pu_{y-\max}$ | Task Space Front Edge | 400 | m |
| $pu_{z-\min}$ | Task Space Bottom Edge | 30 | m | $pu_{z-\max}$ | Task Space Top Edge | 150 | m |
| $d_{\text{end}}$ | Advance End RTH Distance | 30 | m | $V_{x-\max}$ | $x$ Dire. Maximum Velocity | 10 | m/s |
| $V_{y-\max}$ | $y$ Dire. Maximum Velocity | 10 | m/s | $V_{z-\max}$ | $z$ Dire. Maximum Velocity | 5 | m/s |
| $T_{f_{\text{end}}}$ | Maximum COL Time | 500 | s | $T_{r_{\text{end}}}$ | Maximum RTH Time | 100 | s |
| $n_{\text{UAV}}$ | UAV Number | 4 | - | $n_{\text{WS}}$ | WS Number | 400 | - |
| $\text{AoI}_{\max}$ | Maximum AoI of WSs | 0.8 | - | $BC$ | Battery Capacity | 155520 | J |
| $Pw_{\text{cmp}}$ | Computation Power | 20 | W | $T_{\text{update}_{\text{AoI}}}$ | AoI Updating Time | 40,50,60 | s |
| $Pw_{ur}$ | UAV Received Power | 30 | dBm | $m_{\text{UAV}}$ | UAV Mass | 0.2 | kg |
| $g$ | Gravitational Acceleration | 9.8 | - | $\rho_{\text{air}}$ | Air Density | 1.225 | kg/m$^3$ |
| $v_{\text{th}}$ | Hovering Speed Threshold | 0.1 | m/s | $C_d$ | Viscosity Coefficient | 0.5 | $-$ |
| $n_{\text{prp}}$ | Propeller Number | 4 | - | $R_{\text{prp}}$ | Propeller Radius | 0.1 | m |
| $\eta$ | Mechanical Efficiency | 0.8 | - | $A_{\text{surf}}$ | UAV Fuselage Area | 0.01 | m$^2$ |
| $f_c$ | Signal Frequency | 2.8 | GHz | $d_{\text{WS}}$ | Distance between WSs | 20 | m |

Table 6: Hyperparameters for the Agricultural Application.

| Symbol | Definition | Value | Symbol | Definition | Value |
|---|---|---|---|---|---|
| $\alpha_1$ | AoI Weight Parameter in Collection Task | 2 | $\alpha_2$ | Penalty for visiting same grids | 0.5 |
| $\alpha_3$ | AoI Weight Parameter in Return Task | 0.1 | $\alpha_4$ | Motivation for Exploration | 0.01 |
| $\alpha_5$ | Weight Parameter for flying out | 10 | $\gamma$ | Discounted Factor | 0.99 |
| $r_a$ | Learning Rate for Actor | $10^{-4}$ | $r_{ct}$ | Learning Rate for Critic | $10^{-5}$ |
| $r_b$ | Learining rate for BPN | $10^{-5}$ | $\mathcal{B}_{\mathcal{P}}$ | PAN Training Batch Size | 512 |
| $\mathcal{D}$ | Replay Buffer Size | $2^{16}$ | $\mathcal{B}$ | RL Training Batch Size | 128 |
| $epi_{\max}$ | Maximum Training Episode | 10000 | $epo_{\max}$ | PAN Pre-training Maximum Epoch | 1000 |
| $\xi$ | Soft Update parameter | 0.005 | $\tilde{S}$ | Scenario Number | 10 |
| $r_G$ | Learning Rate for GNN | $10^{-3}$ | $r_P$ | Learning Rate for PAN | $10^{-4}$ |
| $\beta_P$ | PAN Normalization Coefficient | 0.01 | $\beta_G$ | GNN Normalization Coefficient | 0.01 |

