# OpenReview forum: "FM-EAC: Feature Model-based Enhanced Actor-Critic for Multi-Task Control in Dynamic Environments"
_ICLR.cc/2026/Conference — ICLR 2026 Conference Withdrawn Submission_

### Official Review · Reviewer_2ZXe · 2025-10-17

**Soundness:** 2
**Presentation:** 2
**Contribution:** 2
**Rating:** 4
**Confidence:** 3

**Summary:**

The paper proposes FM-EAC, a reinforcement learning algorithm that combines model-based and model-free approaches for multi-task UAV control in dynamic environments. The key claimed contributions are: (1) a feature model that extracts environmental representations using sub-networks (GNN, PAN, BPN) to improve generalizability across scenarios, (2) an enhanced actor-critic framework that decouples actors and critics for handling concurrent and sequential tasks, and (3) experimental validation in urban (package delivery + MEC) and agricultural (data collection + charging) applications showing improved performance over baseline methods.

**Strengths:**

1. The paper addresses a practically relevant problem of multi-task UAV control in dynamic environments with real-world applications.

2. The modular feature model design allowing customizable sub-networks (GNN, PAN, BPN) for different scenarios is a useful architectural choice.

3. The paper provides detailed mathematical formulations and algorithms for the enhanced actor-critic framework.

**Weaknesses:**

1. Technical Contributions and Novelty. The enhanced actor-critic is a straightforward extension using separate critics for different rewards, which lacks novelty. The paper provides no theoretical justification (convergence, sample complexity) for why feature models improve generalization, and fails to compare against the most relevant meta-RL methods.

2. Experimental Evaluation. Only one MBRL baseline (MBPO) and no meta-RL methods are compared, while the train/test setup is vague ("3-5 out of 10 maps") with no out-of-distribution testing or statistical significance analysis. The ablation study does not isolate individual components, and critical analyses like learning curves and feature visualizations are missing.

3. Presentation and Writing. Figure 1 is cluttered and unclear about data flow and component interactions.

**Questions:**

1. Generalization claims: Can you provide rigorous evaluation with clear train/test splits, out-of-distribution scenarios, multiple random seeds with statistical tests, and comparisons to meta-RL methods?

2. Feature model necessity: Can you show through proper ablation that environmental features improve performance over standard state representations, and visualize what features are actually learned?

3. Baseline comparisons: Why are recent MBRL methods, meta-RL methods, and multi-task RL methods excluded?

4. Computational cost: What are the training time and sample complexity? How many environment interactions are needed for convergence?

---

### Official Review · Reviewer_bU96 · 2025-11-01

**Soundness:** 2
**Presentation:** 2
**Contribution:** 2
**Rating:** 4
**Confidence:** 2

**Summary:**

This paper introduces the FM-EAC algorithm to overcome the limitations of current reinforcement learning methods regarding multi-task control in dynamic environments and poor transferability. FM-EAC merges the strengths of model-based and model-free RL. A key feature is the Feature Model, which uses subnetworks like GNN or PAN to extract robust environmental characteristics, significantly boosting the algorithm's generalizability. The Enhanced Actor-Critic framework decouples the networks, supporting simultaneous policy optimization for both concurrent tasks and sequential tasks. Experiments confirm that FM-EAC consistently outperforms various state-of-the-art algorithms in terms of performance, speed, and stability.

**Strengths:**

This article is written smoothly and summarizes the work done in concise language. Meanwhile, FM-EAC has excellent generalization ability and can be applied in multiple engineering tasks, making significant progress. Additionally, Figure 1 in this article is very beautiful and impressive.

**Weaknesses:**

Lack of display of experimental results. The author applied FM-EAC to unmanned aerial vehicle control tasks and several other tasks in the fields of industrial and agricultural engineering. However, these experiments lack specific implementation details and do not provide image or video displays to aid understanding. This confuses readers and raises doubts about the authenticity of the article.

**Questions:**

1. In the ablation experiment, the author replaced EAC with the original Actor Critic and obtained GNN-OAC and PAN-OAC. However, the author also mentions that the structures of EAC and OAC are the same in agricultural applications. Why? Additionally, NFM-OAC should also be included in ablation experiments.
2. Can the author provide video presentations of various experiments? This is crucial for understanding the true performance of the model.

---

### Official Review · Reviewer_mtSr · 2025-11-01

**Soundness:** 2
**Presentation:** 2
**Contribution:** 1
**Rating:** 2
**Confidence:** 4

**Summary:**

This paper proposes FM-EAC (Feature Model-based Enhanced Actor-Critic), an algorithm that integrates aspects of MFRL and MBRL for multi-task control in dynamic environments. The authors claim that they combine a "feature model" of the environment inspired by MBRL with an "enhanced actor-critic", which decouples the networks across tasks. Experiments are conducted in two simulated UAV applications - urban delivery and agricultural monitoring. The proposed approach is compared against standard RL algorithms such as SAC, TD3, MBPO, etc. While the paper addresses an application area of potential interest, it is too application-specific; the technical novelty and methodological rigor are limited.

**Strengths:**

1. **Interesting application domain** - the focus on UAV-based multi-task control is practically relevant and demonstrates potential impact for real-world RL deployment.
2. **Broad contextualization** - the paper cites a wide range of MFRL and MBRL baselines

**Weaknesses:**

1. **Lack of novelty and conceptual depth** - despite the title, FM-EAC is effectively a standard actor-critic (close to SAC/TD3) with auxiliary feature-extraction modules. There is no genuine model-based component (no learned dynamics model, planning step, or synthetic rollout), contradicting the model-based claim. The "enhanced" actor-critic merely duplicates critics for primary/secondary tasks, which is a known trick from multi-objective and multi-head architectures.
2. **Pedagogical rather than research-grade presentation** - Sections 1 and 2 read like an undergraduate tutorial on RL. Large parts restate textbook material from Sutton & Barto (2018) without adding insight. Much of Section 3 expands routine SAC equations in unconventional notation, obscuring rather than clarifying the method.
3. **Factual inaccuracies and unsupported claims** - the introduction suggests that MBRL methods lack value functions, which is incorrect. The paper equates adding auxiliary features to "model-based" reasoning, which misrepresents the MBRL definition.
4. **Methodological shortcomings**.
    * Experiments are limited to only two tasks, both within narrow UAV simulations. No generalization across fundamentally different domains is shown.
    * There is no evidence of statistical robustness. The number of random seeds is unspecified, yet tables report +- values without explanation
    * The “ordinary actor-critic” baseline used in ablation (Sec. 4.3) performs implausibly poorly, suggesting either mis-tuning or inconsistent comparison.
    * The feature-model ablation does not include standard baselines such as SAC + auxiliary GNN features, which would isolate the effect of the architecture.
5. **Writing and clarity** - the text is verbose and repetitive and uses non-standard symbols throughout. e.g. Figure 1 is extremely cluttered and not easily interpretable. No need to visualize what a neural network looks like. The use of the feature models is unclear from just the figure.

**Questions:**

1. How many random seeds were used per experiment? What do the reported +- values represent - standard deviation or standard error?
2. In the ablation (Sec. 4.3), what precisely constitutes the “ordinary actor-critic” baseline? Is it identical to SAC/TD3 with equivalent hyperparameters?
3. Can you clarify in what sense FM-EAC is model-based? Is there any explicit learned dynamics model or planning component?
4. How are the feature-extraction sub-networks (GNN/PAN/BPN) trained relative to the policy? Jointly, alternately, or frozen?

---

### Note · Authors · 2025-11-20

I have read and agree with the venue's withdrawal policy on behalf of myself and my co-authors.